# DrugSniper, a Tool to Exploit Loss-Of-Function Screens, Identifies *CREBBP* as a Predictive Biomarker of VOLASERTIB in Small Cell Lung Carcinoma (SCLC)

**DOI:** 10.3390/cancers12071824

**Published:** 2020-07-07

**Authors:** Fernando Carazo, Cristina Bértolo, Carlos Castilla, Xabier Cendoya, Lucía Campuzano, Diego Serrano, Marian Gimeno, Francisco J. Planes, Ruben Pio, Luis M. Montuenga, Angel Rubio

**Affiliations:** 1Department of Biomedical Engineering and Sciences. School of Engineering, University of Navarra, 20018 San Sebastián, Spain; fcarazo@tecnun.es (F.C.); ccastilla.1@tecnun.es (C.C.); xcendoya@tecnun.es (X.C.); mgimenoc@unav.es (M.G.); fplanes@tecnun.es (F.J.P.); 2Program in Solid Tumors, Center for Applied Medical Research (CIMA), CIBERONC and Navarra’s Health Research Institute (IDISNA), 31008 Pamplona, Spain; cbertolo@unav.es (C.B.); dserrano@unav.es (D.S.); lmontuenga@unav.es (L.M.M.); 3University of Luxembourg, 4365 Esch-sur-Alzette, Luxembourg; lucia.campuzano.delrio@gmail.com; 4Department of Pathology, Anatomy, and Physiology, School of Medicine, University of Navarra, 31008 Pamplona, Spain; 5Department of Biochemistry and Genetics, School of Sciences, University of Navarra, 31008 Pamplona, Spain

**Keywords:** biomarker, small cell lung carcinoma, SCLC, CREBBP, PLK1, gene essentiality, drug, treatment, DEMETER score, RNAi, precision medicine, drug repositioning

## Abstract

The development of predictive biomarkers of response to targeted therapies is an unmet clinical need for many antitumoral agents. Recent genome-wide loss-of-function screens, such as RNA interference (RNAi) and CRISPR-Cas9 libraries, are an unprecedented resource to identify novel drug targets, reposition drugs and associate predictive biomarkers in the context of precision oncology. In this work, we have developed and validated a large-scale bioinformatics tool named DrugSniper, which exploits loss-of-function experiments to model the sensitivity of 6237 inhibitors and predict their corresponding biomarkers of sensitivity in 30 tumor types. Applying DrugSniper to small cell lung cancer (SCLC), we identified genes extensively explored in SCLC, such as Aurora kinases or epigenetic agents. Interestingly, the analysis suggested a remarkable vulnerability to polo-like kinase 1 (*PLK1*) inhibition in *CREBBP*-mutant SCLC cells. We validated this association in vitro using four mutated and four wild-type SCLC cell lines and two *PLK1* inhibitors (Volasertib and BI2536), confirming that the effect of *PLK1* inhibitors depended on the mutational status of *CREBBP*. Besides, DrugSniper was validated in-silico with several known clinically-used treatments, including the sensitivity of Tyrosine Kinase Inhibitors (TKIs) and Vemurafenib to *FLT3* and *BRAF* mutant cells, respectively. These findings show the potential of genome-wide loss-of-function screens to identify new personalized therapeutic hypotheses in SCLC and potentially in other tumors, which is a valuable starting point for further drug development and drug repositioning projects.

## 1. Introduction

In recent years, genetic alterations, such as DNA mutations, translocations, or copy number variations, have been used as a source of therapeutic targets and therapy response biomarkers in cancer [1]. However, for certain tumor types, targeted treatments have not yet been discovered or associated with proper biomarkers of response. Additionally, many drug development projects and clinical trials fail because of the lack of proper biomarkers to correctly select sensitive subpopulations of patients [2,3,4].

Analyzing genetic essentiality in specific aberrant phenotypes has been demonstrated as an unprecedented approach to identify drug targets and develop personalized therapies in cancer. This is the case for the sensitivity to *HER2*-inhibitors for tumors with *ERBB2* amplification in breast cancer, the vemurafenib sensitivity of *BRAF*-mutant cells in melanoma, or the relevance of *EGFR* mutation for *EGFR*-inhibitor treatments in non-small cell lung cancer (NSCLC). This approach is especially relevant in tumors that still lack a clear treatment guideline, such as small cell lung carcinoma (SCLC).

SCLC, accounting for around 15% of all lung cancers, is an aggressive neuroendocrine malignancy characterized by a high proliferation rate and early development of widespread metastases. These features contribute to the extremely poor prognosis with a median survival of 7 months after patients are diagnosed [5]. Most SCLC tumors are initially chemosensitive, however, acquired resistance development and cancer recurrence occur often and rapidly [6], revealing the need for new therapeutic approaches. In recent years, promising molecular druggable pathways have been associated with certain subtypes of SCLC based on different mutational and transcriptional patterns [7]. Unfortunately, no targeted therapy has yet demonstrated sufficient efficacy to be considered for routine treatment of SCLC, which is likely due to the lack of knowledge of predictive biomarkers of tumor sensitivity to such emerging targeted drugs. Some examples of promising targets that lack robust companion biomarkers in SCLC are those related to inhibitors of polo-like kinase 1 (*PLK1*) [3,4] or cyclin-dependent kinases (*CDKs*) [2]. In this context, computational technologies could be of great value for the identification and successful application of biomarker-guided targeted therapies.

Large-scale loss-of-function screens, such as RNA interference (RNAi) and CRISPR-based libraries, have been used to delineate novel potential drug targets and predictive biomarkers in human cancer cells. In this context, the Project Achilles [8,9,10], the Project DRIVE [11] and the Project Score [12] performed genome-wide gene-knockouts aiming at establishing relevant essential genes in large sets of cancer cell lines, most of them are collected in the Cancer Cell Line Encyclopedia [13]. Several studies have successfully used these data in combination with other omics, mainly mutations, expression and copy number variations and to predict sensitivity to gene silencing [8,9,14,15]. However, little work has been conducted to relate this resource to define personalized treatments and to predict drug sensitivity biomarkers.

In this research, we present the DrugSniper tool, a bioinformatics software to identify response biomarkers for targeted therapies in 30 tumor types for which cell lines have been derived and included in the Cancer Cell Line Encyclopedia (CCLE). To this aim, DrugSniper modeled the sensitivity of 6237 protein-targeting drugs with the essentiality scores of their corresponding targets for 412 cancer cell lines. In the present work, we will refer to a specific study carried out on SCLC cell lines using DrugSniper.

By integrating protein-targeted drug information with the mutational landscape of SCLC cells, we identified genes extensively explored in SCLC as potential targets, such as Aurora kinases or epigenetic agents. Interestingly, the analysis suggested a remarkable vulnerability to polo-like kinase 1 (*PLK1*) inhibition in *CREBBP*-mutant SCLC cells. We validated this association in vitro using four *CREBBP* mutated and four wild-type SCLC cell lines and two *PLK1* inhibitors (Volasertib and BI2536), confirming that the effect of *PLK1* inhibitors depended on the mutational status of *CREBBP*. Besides, we further validated DrugSniper in-silico in tumor types different from SCLC, focusing on several known personalized treatments, including the sensitivity of Olaparib in *BRCA* mutant patients and to Vemurafenib in *BRAF* mutant cells.

DrugSniper could facilitate the stratification of patients based on the presence of biomarkers that predict the response to current therapies, as well as the repositioning of drugs based on the presence of specific biomarkers in SCLC and other tumors. This approach will thereby open new avenues for personalized therapies, potentially improving outcomes of patients diagnosed with cancer from multiple origins.

## 2. Results

### 2.1. DrugSniper: A Computational Pipeline to Predict Putative Mutation Biomarkers of Targeted Drug Sensitivity

Developing a successful precision medicine strategy requires two main elements: the therapy itself (e.g., a drug or other clinical intervention) and a predictive biomarker, which indicates whether the treatment will be effective in a patient or not. In this study, we developed an application (DrugSniper) that models drug sensitivity by the essentiality scores of genes associated with protein targets (therapy) and predicts mutations (predictive biomarkers) that discriminate resistant and sensitive cells. In total, the current version of DrugSniper includes 6237 compounds and 17,098 targets in 412 cancer cell lines.

DrugSniper’s functionality facilitates the identification and selection of precision medicine therapies in four real-life scenarios (Figure 1A). In the first scenario, the researcher may have a set of putative protein targets related to the cancer type under study and wants to find gene mutations that predict their essentiality, as well as available drugs targeted to them. This functionality of DrugSniper is useful to compare possible targets and to stratify cancer patients in drug development projects. In the second situation, the researcher may have a list of drugs and wants to identify gene mutations that are good predictive biomarkers for such drugs in a specific tumor type. In the third scenario, the researcher may have one or more mutations of interest and wants to identify targets/drugs whose essentiality depends on the status of the input mutations. Finally, in the fourth case, the researcher may use the tool with no prior hypotheses by selecting “all protein targets, all drugs and all mutations”. This option outputs a ranking of targets/drugs suitable for a cohort of cells of the tumor under study, with their best corresponding predictive mutation biomarkers (Figure 1B). In this case, therefore, the tumor histological subtypes—or a specific cohort of cell lines—are the only input data required.

DrugSniper integrates three sources of large-scale datasets: targeted drug information, genome-wide RNAi libraries and mutational profiles (Figure 2A). We included both approved and investigational drugs targeted to protein inhibition. To do so, we retrieved the data publicly available in the ChEMBL [16] and DrugBank [17] online repositories. We started by searching the ChEMBL compound list for annotated mechanisms of action and indications and selected inhibitors of human protein targets, leaving us with 925 compounds and 456 protein targets. This resulted in 2126 drug-target interactions (DTIs) and 4106 drug indications. The same search was performed in DrugBank, obtaining 634 drugs with 433 protein targets inhibited by them in 1339 DTIs. Additionally, we expanded this list with high potency inhibitory DTIs contained in ChEMBL assays, by selecting only those whose IC_50_ or K_i_ values were below 1 µM. Integrating the three lists of drugs and targets, we were left with 6237 compounds that constituted DTIs with 2284 protein targets.

Project Achilles measures how individual genes affect cell survival through RNAi experiments. The DEMETER score [9,18] is a statistical summarization of the Project Achilles data that quantifies essentiality scores for 17,098 genes. This score minimizes the off-target effects and outperforms other scores, such as the ATARiS score [8] or Bayes factors [19]. The DEMETER score is negatively correlated with the essentiality of a gene. Creators of DEMETER suggested a cutoff of −2 in the DEMETER score to establish the limit of cell viability, i.e., genes with a DEMETER score lower than this threshold can be considered essential for a cell line. On the other hand, the CCLE includes genomic information of cancer cell lines, such as DNA mutations or gene expression. We downloaded the mutational profiles of 1660 genes, maintaining the filters used by CCLE on the mutations to avoid common polymorphisms, low allelic fractions, putative neutral variants, and mutations outside the coding sequences for all transcripts.

We integrated mutational profiles with DEMETER scores for 412 cell lines and developed a statistical pipeline to identify gene mutations as predictive biomarkers for targeted therapies (Figure 2B,C). The statistical model is based on the Independent Hypothesis Weighting procedure [20] and limma [21] to state the probability of the sensitivity to a gene knockdown to be differential in mutant and wild-type cell lines. See the methods section for more details of the pipeline and a quick start for the application inAppendix A.

### 2.2. Validation with Previously Known Associations

As a proof of concept validation of DrugSniper, we focused on essential gene-predictive mutation pairs that were already in clinical use and checked their statistical significance in DrugSniper. Interestingly, DrugSniper identified several of these known drugs and biomarkers that define current clinical guidelines.

Acute Myeloid Leukemia (AML) is characterized by the absence of associated treatments. However, Tyrosine Kinase Inhibitors (TKIs) were recently prescribed as a possible therapy for *FLT3*-mutated AML [22]. DrugSniper was able to predict the importance of different TKIs drugs in *FLT3*-mutant AML cell lines (*p*-value = 2.54 × 10^−4^ and local false discovery rate (lfdr) = 0.18). Another example is Vemurafenib, a drug widely used to treat tumors driven by pathogenic variants in the *BRAF* gene, e.g., melanoma or skin cancers. This drug inhibits *BRAF* activity when *BRAF* is altered [23]. DrugSniper predicted *BRAF* vulnerability when *BRAF* was mutated in skin cancer (*p*-value = 2.59 × 10^−3^). Likewise, Olaparib is a drug that exploits synthetic lethality with successfully proven efficacy [24]. This drug inhibits *PARP* polymerases that are essential for DNA repair in the presence of mutated *BRCA* genes [25]. DrugSniper predicted the essentiality of *PARP* in tumors with *BRCA2* mutant variants (*p*-value = 6.62 × 10^−3^ and local FDR = 0.1). Similarly, recent publications showed the efficacy of using Trametinib (*MEK* inhibitor) in lung cancer cells with *ATM* mutations [26]. *MEK* is a gene family that contains several mitogen-activated protein kinases (*MAPK*). DrugSniper found *MAPK7*—a *MEK* family gene—to be essential in lung cancer when *ATM* was altered (*p*-value = 5.99 × 10^−4^

An essential therapy for *EGFR*-mutant tumors is Erlotinib. This drug is frequently used to treat non-small cell lung cancer but could be used to treat different *EGFR*-mutant cancers [27]. *EGFR* becomes essential when *EGFR* is mutated in cancer cells because of oncogene addiction [28]. DrugSniper predicted this mechanism in lung, esophagus, pleura and prostate cancer cell lines (*p*-value = 2.74 × 10^−2^ and local FDR = 0.03).

Remarkably, Drugsniper is capable of identifying not only loss-of-function but also gain-of-function mutations as predictive biomarkers of genetic essentiality, as shown in a supplementary case study using lung adenocarcinoma (Appendix A). Specifically, DrugSniper identifies 1030 target-biomarker pairs that are significantly relevant in lung adenocarcinoma, 38 out of which have a biological interaction annotated in the String database. Interestingly, the ranking includes *KRAS*–*KRAS*mut pair (top 1 pair when filtering by String), which is a well-known oncogene addiction [28], triggered by a gain-of-functionality of *KRAS* due to its activating mutation. More examples of associations between clinically used drugs and biomarkers found by DrugSniper are collected in Appendix A.

It is important to note that the pairs: *PARP1*–*BRCA2*, *MAP3K7*–*ATM* and *EGFR*–*EGFR* were validated in the tab “Visualize case-by-case” focusing only on the statistical significance between such pairs. This means that they do not appear in the current pipeline “Predict Biomarkers for a Target Gene” due to the stringent filters applied (e.g., DEMETER Essentiality cutoff equal to −2 and Delta Essentiality bigger than −2). We set these filters to minimize the false positives when screening, which is crucial for wet lab hypothesis validation. Therefore, these filters eliminated some of the well-known and clinically used pairs mentioned above, although they were statistically significant when assessed as pairs (Appendix A.

### 2.3. CREBBP Mutation is a Predictive Biomarker for PLK1 Inhibitors Efficacy in Small Cell Lung Cancer Cell Lines

We applied DrugSniper to SCLC cell lines (*n* = 22) to evaluate the potential application of this approach in a blinded situation. We considered the first of the scenarios mentioned above, i.e., we did not have a priori selected drugs or targets to test in SCLC. The goal was to obtain a list of putative drugs/targets with their corresponding predictive mutation biomarkers for this elusive and aggressive cancer type. In this case, the tool was first used to explore SCLC cell viability for all available gene silencing experiments. The filters used to select the first list of essential genes were:
To be essential in >20% of SCLC cell lines, with a threshold of −2 for the DEMETER score for essentiality. Using this filter, we require that at least 20% of SCLC samples die when genes considered “essential” are knocked down.To be specific for SCLC cell lines with an odds ratio >1 (i.e., the percentage of cell lines in which the selected genes were essential should be larger for SCLC than for cell lines derived from any other tumor types). This specificity parameter states that at most for 20% of the other cell-lines, the selected gene is essential. Henceforth, the gene is hypothesized to be a good drug target specifically for SCLC.To have a minimal expression score of 1 TPM (transcripts per million) in more than 75% of the SCLC cell lines.

Using these three criteria, 277 targets were initially found to be essential and specific to SCLC cell lines (Appendix A). These initial targets were ranked to predict the best combination of target-response biomarkers. For the final list of hypotheses (Table 1), we required that the members of each pair of target and response biomarkers were also functionally/biologically related in the STRING protein–protein interactions database (an option also implemented in DrugSniper).

Using the above-mentioned filters, DrugSniper predicted 28 putative protein targets with their corresponding predictive mutation biomarkers (Appendix A). Interestingly, several top-ranked targets belong to important pathways known to be dysregulated in SCLC. Some examples are *LSD1/KDM1A*, which catalyzes the demethylation of lysines 4 and 9 in histone 3 (H3K4 and H3K9), is involved in the development and tissue-specific differentiation [29] and is a known target with an experimental drug being tested in SCLC [30]; *CASP8AP2/FLASH*, which supports cancer cells’ epithelial-to-mesenchymal (EMT) transition and inactivates Notch signaling [31,32]; *SMARCA4/BRG1*, a component of the SWI/SNF-B (PBAF) chromatin remodeling complex, which activates neuroendocrine transcription and has been associated with relevant regulating roles in SCLC [33]; or *PLK1,* a serine/threonine-protein kinase that is a key regulator of mitotic progression and is also required for the spindle assembly checkpoint [34]*. PLK1* has already been proposed as a potential target in SCLC and other tumors. A Phase II trial with BI2536, a *PLK1* inhibitor, has been carried out, although it did not show efficacy in the treatment of relapsed SCLC [3,4].

To validate the accuracy of these results, we selected the top hypothesis: *PLK1* inhibition with the mutation of *CREBBP* as a response biomarker (∆Ess = −3.9, *p*-value = 1.45 × 10^−3^, lfdr = 0.15; Table 1 and Figure 1B). Remarkably, *CREBBP* lof mutations in SCLC showed to have no co-occurrence with other gene variants, which makes *CREBBP* a potential biomarker (Appendix A). The presence of *CREBBP* as a predictive biomarker of multiple essential genes could be counterintuitive. Interestingly, when performing a pathway enrichment analysis of genes related to *CREBBP* mutation, we discovered that a significant part of the genes shared similar pathways (STRING’s PPI enrichment *p*-value: 0.000163), and also the KEGG pathways spliceosome (false discovery rate (FDR) = 1.62 × 10^−5^), ribosome (FDR = 0.0049) or cell cycle (FDR = 0.0049).

*PLK1* has several experimental inhibitory drugs, so we were able to validate the prediction derived from DrugSniper by exploring the efficacy of two commercially available *PLK1* inhibitors, Volasertib and BI2536, over a panel of SCLC cells in a cell viability (MTS) assay. We compared the relative cell growth after 72 h of treatment with serial dilutions of *PLK1* inhibitors and analyzed the statistical results using the R library drc [35].

The dose–response curves for Volasertib revealed IC_50_ values significantly higher for the non-mutated SCLC cell lines NCI-H841, NCI-H889, NCI-H2171 and NCI-H146 than for the *CREBBP* mutated SCLC cell lines NCI-H1048, NCI-H1963, NCI-H211 and HCC33 (13.3 vs. 3.7 nM, respectively; *p* = 1.35 × 10^−12^) (Figure 3A). Very similar data were obtained after treatment with BI2536. The IC_50_ value was significantly higher for wild-type than for mutant cell lines (10.5 vs. 2.3 nM, respectively; *p* = 2.76 × 10^−14^). IC50 values for each cell line can be found in the Appendix A.

Sensitivity to *PLK1* inhibitors was also measured using a colony formation assay. The *CREBBP* mutated (*CREBBP*-MUT) cell line NCI-H1048 reduced its ability to form colonies significantly after exposure to 2.5 nM of Volasertib (*p* = 7.24 × 10^−12^) and 5 nM of BI2536 (*p* = 9.37 × 10^−7^), showing a significant global effect of both inhibitors (*p* = 4.84 × 10^−14^). In contrast, the *CREBBP* wild-type (*CREBBP*-WT) cell line NCI-H841 was more resistant to Volasertib or BI2536 treatment (Figure 3B,C). Additionally, we conducted a cell cycle analysis by FACS after 24 h of treatment with 5 nM of Volasertib or BI2536 in NCI-H841 and NCI-H1048 cells (Figure 3D). There was a minor cell cycle G2M arrest after Volasertib exposure in the *CREBBP*-WT NCI-H841 cells. In contrast, *CREBBP*-MUT cells NCI-H1048 showed a completely disrupted cell cycle after both Volasertib and BI2536 treatment at low concentrations. The arrest became more significant after BI2536 treatment. These experiments strongly support the hypothesis that SCLC cell viability is significantly different depending on the mutational status of the *CREBBP* gene.

## 3. Discussion

Present trends of targeted drug development projects include the research of companion biomarkers of sensitivity. The absence of predictive biomarkers explains why some promising clinical trials fall by the wayside and do not change the current conventional treatments [36]. Finding robust biomarkers for cancer therapies would recover drugs even when they are indicated only for a low percentage of patients.

In this context, we investigated how to discover novel targets and associated predictive biomarkers by using recent large-scale loss-of-function experiments, such as RNAi or CRISPR-Cas9 screens. We have developed DrugSniper, a pioneer computational approach that provides working hypotheses to develop personalized therapies based on the status of gene mutations.

DrugSniper has been developed using genome-wide loss-of-function screenings. However, it could be argued that using drug panels, such as IC_50_ screenings, is a better model for treatment efficacy. Nevertheless, IC_50_ experiments are known to have high variability: they may return disparate results even for the same drugs and cell lines [37]. We corroborated that two main IC_50_ databases, namely GDSC [38] and CCLE [13], have low consistency when considering the effect of the same drugs in the same cells (Spearman correlation: 0.03 ± 0.12). Besides, the number of drugs included in these experiments is still small (<100 compounds in GDSC and 24 in CCLE) compared with the 6237 compounds that we integrated into DrugSniper. On the other hand, previous works have successfully used genome-wide RNAi screens to model therapy resistance of specific drugs [39]. Genome-wide loss-of-function experiments are, therefore, an encouraging alternative to model target sensitivity.

We applied this tool to SCLC, as it is a tumor with an urgent unmet medical need. DrugSniper predicted a range of drug targets and their corresponding companion biomarkers of sensitivity. Within the top-ranked discoveries, DrugSniper suggested a remarkable vulnerability to *PLK1* inhibition in *CREBBP*-mutant SCLC cells. These findings were validated in vitro using two *PLK1* inhibitors (Volasertib and BI2536), confirming that cells were more sensitive or resistant to both treatments according to our computational predictions. These results and the recall of multiple clinically-used treatments, such as Olaparib or Vemurafenib, suggest the validity of the methodology.

These findings are especially relevant in SCLC for several reasons. On the one hand, *PLK1* plays an essential role in several oncogenic functions including regulation of mitosis and cytokinesis, modulation of genomic stability, induction of cell survival and regulation of cell division [40]. Interestingly, this gene is up-regulated in a variety of human tumors and its expression often correlates with poor prognosis in cancer patients [40]. *PLK1* has long been considered an effective target for anti-mitotic agents and has been the subject of an extensive effort for anti-cancer drug discovery [41]. On the other hand, *CREBBP*, which is mutated in 10% of patients, is an acetyltransferase that plays a central role in histone acetylation, chromatin stability and transcription [42]. Inactivating mutations in *CREBBP* are frequently found in SCLC tissues [43], where it is suggested to act as a tumor suppressor [44]. Remarkably, *CREBBP* and *PLK1* are involved in regulating the activity of *FOXM1*, a transcription factor that orchestrates the transcription of essential genes for cell cycle progression [45]. Additionally, *PLK1* interacts with *FOXM1* by mediating its phosphorylation on Ser-724 and stimulates its transcriptional activity [46].

Most *PLK1* inhibitors involved in clinical trials have shown no response in different cohorts of lung cancer (both SCLC and NSCLC), so far. Volasertib, which is currently in phase II [3], showed an overall response rate of less than 10% and a Progression-Free Survival of 1.4 months [47]. Although all trials demonstrated that the toxicity of Volasertib was manageable, the overall efficacy in lung cancer patients is lower than expected, which increases the interest of this specific finding. According to our prediction, patients bearing a *CREBBP* mutation could be more sensitive to *PLK1* inhibitors. Indeed, the *PLK1* stratification based on the companion biomarker-mutated *CREBBP* that we report herein constitutes a promising approach to improve SCLC therapies, as shown by the in vitro validation. Likewise, this strategy could be extensible to NSCLC (8% of patients have a *CREBBP* mutated background) and other cancer types.

DrugSniper’s approach allows the identification of drugs and biomarkers in a tumor histological subtype or a specific cohort of cell lines. However, the activity and safety of a drug depend on many other factors (ADME characteristics, pharmacokinetics, etc.), which are not modeled by this pipeline. The output of DrugSniper should be considered as a hypothesis that helps a posterior drug development project, which should be validated and tested.

The applicability of this computational pipeline will increase as more loss-of-function experiments are carried out in a wider range of tumors. So far, around 30 cancer types have genome-wide RNAi experiments available, some of which have only a few screened samples. In this context, DrugSniper opens another encouraging alternative: analyzing not only cells by primary site, but also by other tumor characteristics such as tumor histology. Another interesting future line is including genome-wide inhibition screens of normal samples. These experiments would complement DrugSniper’s approach as they would allow the consideration of target inhibition toxicity in normal tissue. DrugSniper methodology can also be extended to other types of biomarkers (not only mutations), such as copy number variations, expression, splicing or, even, epigenetic footprints. Nevertheless, mutations, or, in general, DNA alterations, can be more easily measured than epigenetics or RNA expression and are, therefore, more useful biomarkers.

Large-scale loss-of-function screens are opening a wide range of opportunities to identify new personalized therapeutic strategies in cancer, which is an interesting starting point for further drug development projects. This work proposes a methodology to analyze these valuable resources and discover new applications that use mutations to help define more efficient clinical decisions in cancer.

## 4. Materials and Methods

### 4.1. Data Sources and Preprocessing

CCLE [13] provides public access to genomic data of more than 1000 cancer cell lines. The transcriptomic profiles of these samples were calculated in a previous study [48] from raw RNA sequencing data using Kallisto [49]. This study uses the Gencode 24 transcriptome (GRCh38) as its reference annotation [50]. This version of the transcriptome contains 199,169 transcripts. Gene expression was summarized using the Transcripts Per Million (TPM) measurement. 

We integrated genome-wide RNAi libraries (17,085 knocked-down genes) of 412 cancer cell lines of the Project Achilles [10] with their corresponding gene mutational profiles (mutations in ~1600 genes) obtained from CCLE [13], Shao et al. [8] and Rouillard et al. [51].

Project Achilles interrogated these 412 cell lines for gene essentiality using RNAi screens. From that basis, the DEMETER score was created to quantify the competitive proliferation of the cell lines and minimize the effect of off-target hybridizations by using a statistical model. We used the DEMETER score as a measure of essentiality. The more negative the DEMETER score is, the more essential a gene is for a given cell line. Authors of DEMETER established a cut-off of −2 in this score as a threshold of essentiality. Genes with a DEMETER score lower than this threshold can be considered essential for a cell line. Missing scores of DEMETER were imputed using the nearest neighbor averaging algorithm [52]. Combining these data with drug information databases and mutational profiles, we developed a statistical pipeline to find predictive biomarkers of targeted drugs (Figure 4).

### 4.2. Statistical Model

Let *t* denote the number of RNAi target genes and *n* denote the number of screened samples. Let E be a t×n matrix of target gene essentiality scores with each element $e_{ij}$ represent the DEMETER score for the RNAi target *i* in sample *j*.

Let *p* denote the number of mutated genes in the same *n* screened samples. Let M be a p×n dichotomized matrix whose element $\;m_{ij}$ denotes whether sample *j* is mutant in gene *i*. The resulting $m_{ij}\;$ elements are as follows: (1)$$m_{ij}=\left\{\begin{array}{c} 1,\;\;\;\;\;if\;mutant\;\;\;\;\;\;\;\;\;\left(MUT\right)\;\;\;\\ 0,\;\;\;\;\;\;\;\;if\;wild-type\;\;\;\;\left(WT\right) \end{array}\right..$$

Let *n’* be a subset of screened samples (columns of E and M matrices) that yields an essentiality matrix E*’* and a mutation matrix M’**.** Let $e{’}_{t}$ be the DEMETER score for the RNAi target *t* for the *n’* cell lines cohort. Let $m{’}_{p}$ be a vector with the mutational status of gene *p* for the n’ cell lines. For each pair of RNAi targets (t) and mutations (*p*), a null hypothesis is defined as
(2)$$H_{0}^{g}:E\left(e{’}_{t}|m{’}_{p}\in MUT\right)=E\left(e{’}_{t}|m{’}_{p}\in WT\right).$$

This null hypothesis is, therefore “the essentiality score of a gene is identical in mutant and wild-type cell lines”. To test this hypothesis, we used a moderated t-test implemented in limma [21]. We applied this test for each RNAi target and all the mutations to obtain the corresponding *p*-values. Dealing with these *p*-values implies the correction of a huge number of multiple hypotheses (more than 20 million hypotheses).

To correct for multiple hypotheses, we followed a methodology similar to the IHW (Independent Hypothesis Weighting) procedure [20], which increases the power of a test by grouping the results using covariates. In our case, we divided the *p*-values corresponding to all tests according to the gene that is mutated in each case. For each group, we computed the local false discovery rate (local FDR), which estimates the probability of the null hypothesis to be true, conditioned by the observed *p*-values [53]. The formula of the local FDR is the following:(3)$$P\left(H_{0}|z\right)=local\;FDR\left(z\right)=\frac{\pi_{0}f_{0}\left(z\right)}{f\left(z\right)}\;,$$
where *z* is the observed *p*-values, π_0_ is the proportion of true null hypotheses estimated from the data, $f_{0}\left(z\right)$ is the empirical null distribution—usually a uniform (0,1) distribution for well-designed tests—and $f\left(z\right)$ is the mixture of the densities of the null and alternative hypotheses also estimated from the data. As stated in [53], “the advantage of the local FDR is its specificity: it provides a measure of belief in a gene, it’s ‘significance’ that depends on its *p*-value, not on its inclusion in a larger set of possible values” as it occurs with previous multiple hypothesis correction methods such as *q*-values or the standard False Discovery Rate. The local FDR and π_0_ were estimated using the Bioconductor’s R Package *q-*value [54].

### 4.3. Cell Culture

The human SCLC cell lines NCI-H841, NCI-H1048, NCI-889, NCI-2171, NCI-146, NCI-H1963, HCC33 and NCI-H211 were obtained from the American Type Culture Collection (Manassas, VA, USA). Cells were grown in RPMI 1640 with 10% FetalClone (Thermo Fisher Scientific, Waltham, MA, USA) at 37 °C with 5% CO_2_ in a humidified incubator, except for NCI-H2171 and NCI-H889 cells, which were grown in a HITES medium (DMEM/F12 supplemented with 1% Glutamax, 100 U/mL penicillin, 100 µg/mL streptomycin, 4 µg/mL hydrocortisone (Sigma, St Louis, MO, USA), 5 ng/mL murine EGF and 1% of an insulin–transferrin–selenium mix (Gibco, Thermo Fisher Scientific, Waltham, MA, USA) with 5% FetalClone. Cell lines were authenticated and routinely tested for mycoplasma.

### 4.4. Cell Viability Assays

Cell viability was determined using the 3-(4,5-dimethyl-thiazol-2yl)-5-(3-carboxymethoxyphenyl)-2-(4-sulfophenyl)-2H-tetrazolium (MTS) reduction assay (Promega, Madison, WI, USA) according to manufacturer specifications. Absorbance was measured at 540 and 690 nm in a SpectroStar nano reader (BMG Labtech, Ortenbergm, Germany). The experiments were performed in sextuplicate and repeated at least three times.

### 4.5. Drug Sensitivity Assay (IC_50_)

*PLK1* inhibitors Volasertib and BI2536 were obtained from SelleckChem. SCLC cells were seeded in 96-well plates at a density of 2.5 × 10^3^ cells/well and incubated in a medium with threefold serial dilutions of *PLK1* inhibitors, starting from 10 µM, for 72 h. After incubation with MTS reagent for 3 h, the absorbance was measured. Absorbance reading from the cells incubated without the inhibitor was used for 100% survival. Data were collected from six technical and three biological replicates for each cell line.

### 4.6. Colony Formation Assay

The human adherent SCLC cells NCI-H841 and NCI-H1048 were used in colony formation assays. Cells were seeded at a density of 500 cells/well in a 6-well plate, treated with 0.1% DMSO or increasing doses (2.5–20 nM) of *PLK1* inhibitors (Volasertib and BI2536) for 72 h. The medium was replaced with a drug-free medium every 72 h for 14 days. Clones were fixed in 4% formaldehyde for 30 min, stained with crystal violet for 5 min, scanned and counted. The clonogenic assays were carried out in triplicate and repeated at least three times.

### 4.7. Cell Cycle Analysis

SCLC cell lines were synchronized in a starving medium with 0.5% FetalClone for 12 h. Then, the medium was replaced with a fresh medium with or without 10 nM of the *PLK1* inhibitors Volasertib or BI2536. After 24 h of treatment, cells were collected by centrifugation, washed with PBS, fixed with 70% ethanol, incubated with 0.5 mg/mL RNase (Sigma-Aldrich, St Louis, MO, USA) at 37 °C for 30 min and stained with propidium iodide (Sigma, St Louis, MO, USA). Cell fluorescence was analyzed by flow cytometry on a FACSCalibur platform fitted with a Cell Quest Pro software package (BD Biosciences, San Jose, CA, USA). Cell cycle results were analyzed with FlowJo v9 software (Tree Star, Ashland, OR, USA).

### 4.8. Statistical Analysis

Data are presented as mean ± SD from 3 or more independent repetitions. Analyses of MTS and clonogenic assays were performed using a quasi-Poisson statistical model. Cell proliferation data were fitted and compared (WT vs. MUT). The half-maximal inhibitory concentration (IC_50_) of each inhibitor was determined by nonlinear regression using the R package drc. The statistical significance was also determined using this package.

### 4.9. Availability of Data and Materials

The source code and databases of DrugSniper can be freely downloaded from the GitLab repository: https://gitlab.com/ccastilla.1/DrugSniper. The tool can be run locally following the instructions at the GitLab repository. Additionally, a web-tool version of DrugSniper is available at http://biotecnun.unav.es/app/DrugSniper, which allows the usage of the app by following a few intuitive steps. DrugSniper has been deployed using RShiny in a Docker container framework to facilitate scalability and reproducibility.

## 5. Conclusion 

Genome-wide loss-of-function screens in combination with other genomics data have the potential to identify novel personalized hypotheses for cancer treatment. This approach opens up a wide range of opportunities to identify new personalized cancer therapeutic strategies, which is a valuable starting point for future drug development and repositioning projects. The DrugSniper platform presented here facilitates the identification of new targets and predictive biomarkers in 30 tumors by following a few simple steps, which may contribute to define better and more efficient clinical treatments in the future (i.e., precision medicine).

## Figures and Tables

**Figure 1 cancers-12-01824-f001:**
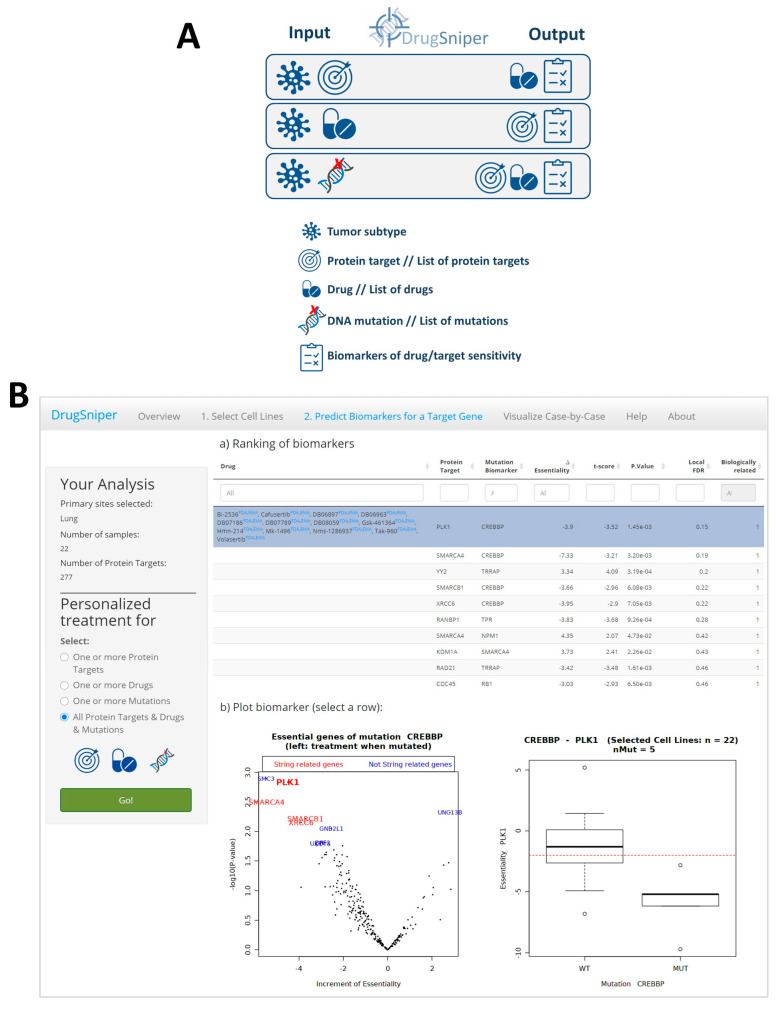
Screenshot of the DrugSniper web application**.** (**A**) Potential applications of the tool. DrugSniper predicts putative drugs and protein targets for a tumor type with their predictive biomarkers for four different inputs: (i) a set of targets and a tumor type, (ii) a set of drugs and a tumor type, (iii) a set of mutations and a tumor type and (iv) only the tumor type. (**B**) Example of the interactive web-application DrugSniper in small cell lung cancer cell lines (*n* = 22). The table shows the ranking of drugs and their protein targets with their corresponding predictive biomarkers. See Table 1 legend to acquire more information about the table’s columns. (Down-left plot) volcano plot showing the highlighted row in the table (*PLK1*-*CREBBP*) in bold red and top essential genes that depend on the mutational status of *CREBBP*. If *CREBBP* is mutated, the *PLK1* gene is essential for small cell lung cancer (SCLC) cells’ viability, suggesting that *CREBBP* is a good predictive biomarker of *PLK1* essentiality. (Down-right plot) box plot showing *PLK1*′s essentiality score (DEMETER score) regarding *CREBBP* mutational status. *PLK1* is essential for *CREBBP*-mutant cell lines. The dotted red line represents the −2 DEMETER score threshold (which is the essentiality threshold proposed by DEMETER’s authors).

**Figure 2 cancers-12-01824-f002:**
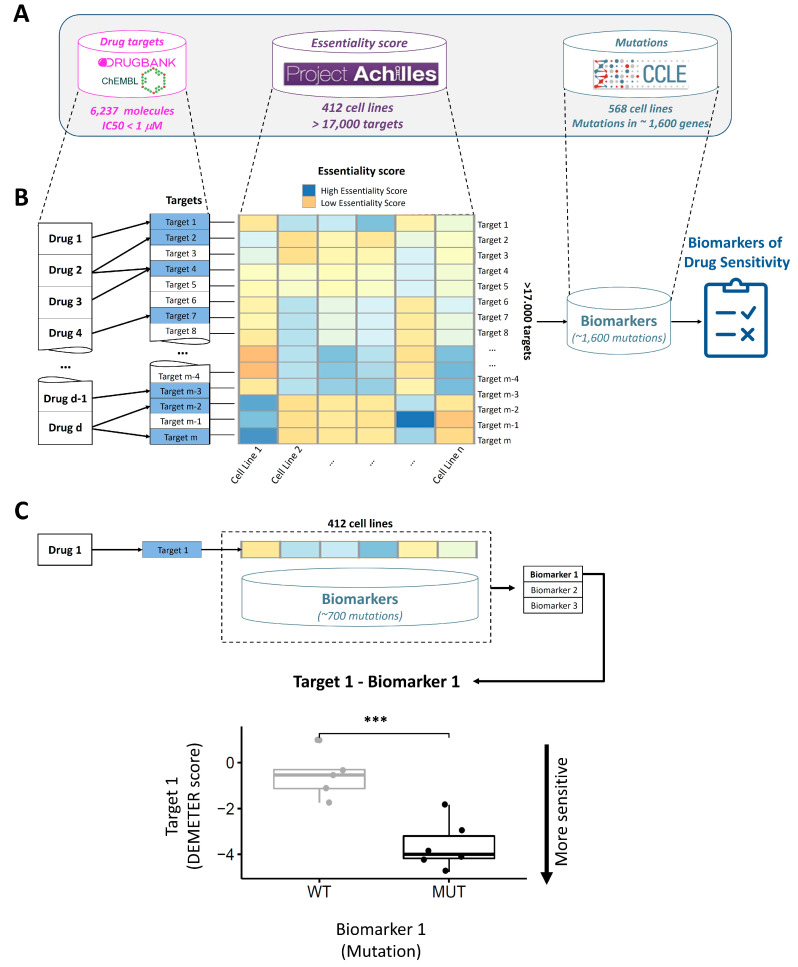
Overview of DrugSniper’s computational pipeline. (**A**) Databases integrated into this work for drugs (6237 molecules), sensitivity scores (17,098 RNA interference (RNAi)) and mutations (mutations in ~1600 genes). (**B**) Schematic summary of DrugSniper’s pipeline to find predictive biomarkers of targeted drugs. Each drug has one or more associated targets, which have an essentiality score for several cell lines. Using a statistical analysis, DrugSniper identifies the best predictive biomarkers of drug targets. (**C**) Example of the pipeline for a single drug. The boxplot shows the sensitivity to the drug target based on the predictive mutation biomarker. The essentiality score (DEMETER score) is negatively correlated with cell viability. In this case, cells with mutations in Biomarker 1 are sensitive to the inhibition of Target 1

**Figure 3 cancers-12-01824-f003:**
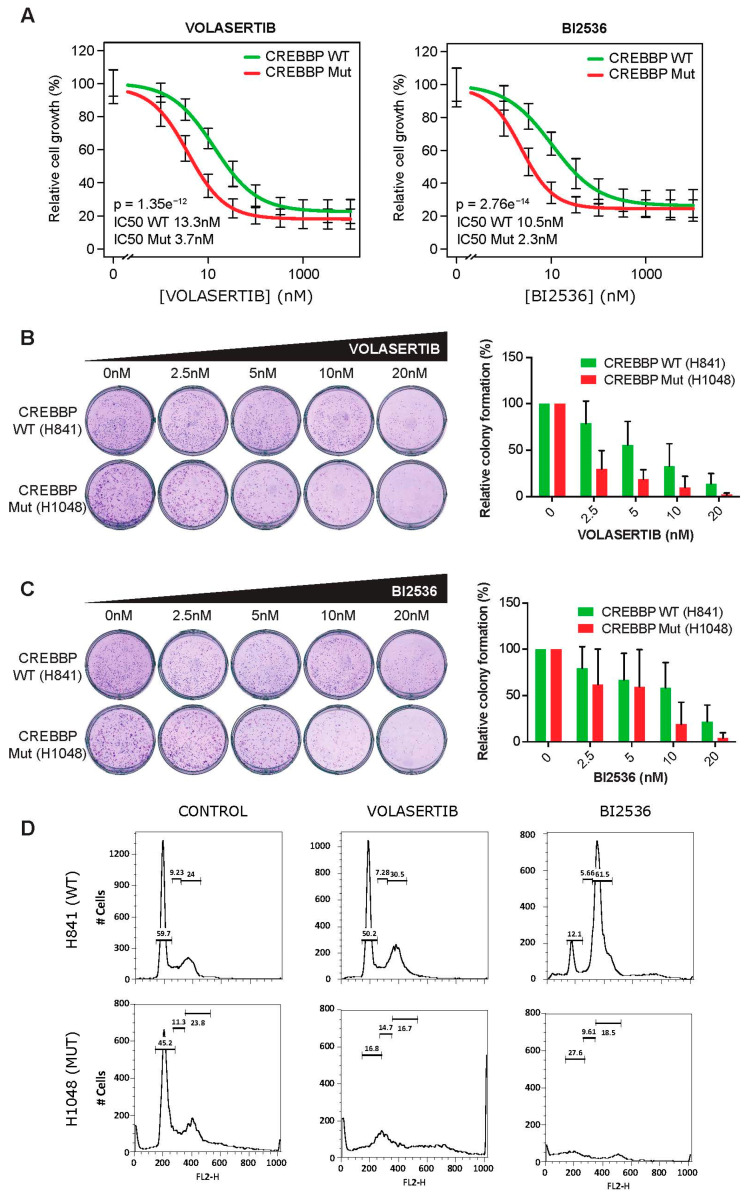
In vitro validation of the sensitivity of *CREBBP*-mutant SCLC cell lines to two *PLK1* inhibitors: Volasertib and BI2536. (**A**) Dose–response curves showing the effect of Volasertib and BI2536 treatment on the viability of *CREBBP*-WT NCI-H841, NCI-H889, NCI-H2171, NCI-H146 cells, and *CREBBP*-MUT NCI-H1048, NCI-H1963, NCI-H211, HCC33 cells. Cells were treated with the indicated doses for 72 h. Cell viability was measured using the cell viability (MTS) assay and the IC_50_ was calculated for each cell line. (**B**) Colony formation assays of NCI-H841 and NCI-H1048 cells. Cells were seeded onto a six-well plate and were treated with vehicle (0.1% DMSO) or increasing doses of Volasertib or BI2536 for 72 h. After treatment, cells were incubated in a drug-free culture medium for 14 days, fixed and stained with crystal violet. (**C**) Quantification of the number of colonies obtained in each condition with Fiji software. (**D**) FACS cell cycle analysis of NCI-H841 and NCI-H1048 cells conducted upon 5 nM Volasertib and BI2536 treatment for 24 h.

**Figure 4 cancers-12-01824-f004:**
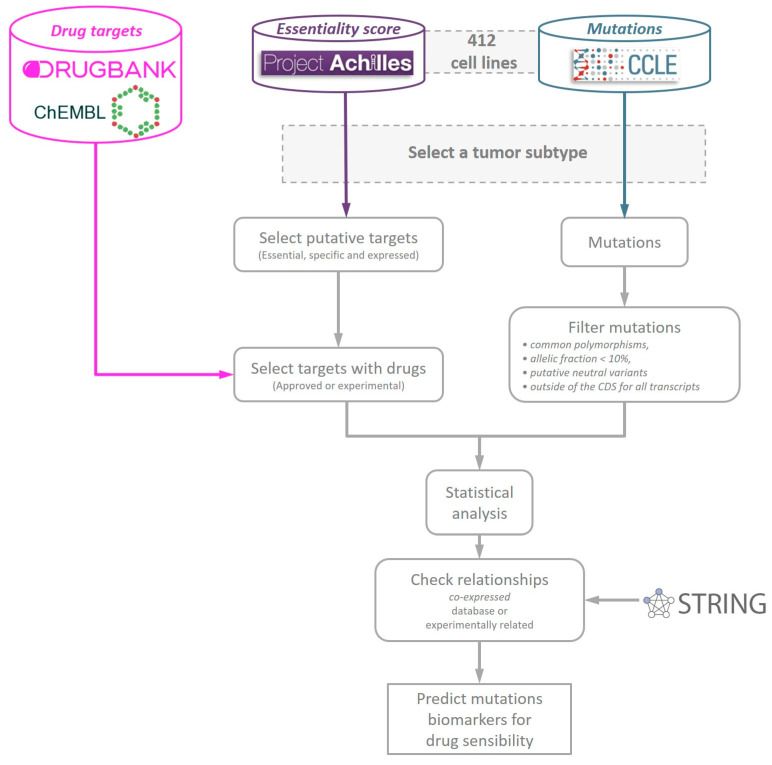
Scheme of the DrugSniper’s computational pipeline. The databases integrated into this work for drugs (DrugBank and ChEMBL), essentiality scores (Project Achilles: DEMETER) and mutations (Cancer Cell Line Encyclopedia (CCLE)).

**Table 1 cancers-12-01824-t001:** Ranking of drug targets and associated predictive mutation biomarkers for small cell lung cancer (SCLC) cell lines using DrugSniper.

Drug Target	Mutation Biomarker	∆Ess	% Mut Patient	*p*-Value	lfdr (adj. *p*-Value)
PLK1*	CREBBP	−3.9	10	1.45 × 10^−3^	0.15
SMARCA4	CREBBP	−7.33	10	3.20 × 10^−3^	0.19
YY2	TRRAP	3.34	4.55	3.19 × 10^−4^	0.2
SMARCB1	CREBBP	−3.66	10	6.08 × 10^−3^	0.22
XRCC6	CREBBP	−3.95	10	7.05 × 10^−3^	0.22
RANBP1	TPR	−3.83	5.45	9.26 × 10^−4^	0.28
SMARCA4	NPM1	4.35	NA	4.73 × 10^−2^	0.42
KDM1A	SMARCA4	3.73	4.55	2.26 × 10^−2^	0.43
RAD21	TRRAP	−3.42	4.55	1.61 × 10^−3^	0.46
CDC45	RB1	−3.03	78.18	6.50 × 10^−3^	0.46
UBE2I	RB1	3.47	78.18	8.67 × 10^−3^	0.47
CDK2*	EPHA5	−3.53	10.91	1.26 × 10^−2^	0.47
ZNF548	PRKG1	−2.82	1.82	4.29 × 10^−2^	0.48
RPS6	TTBK1	3.46	4.55	3.82 × 10^−2^	0.49

The ranking is sorted according to the local false discovery rate (lfdr) (adjusted *p*-value), more information in the Methods Section. The column ∆Ess represents the average change in the DEMETER score between mutated and wild-type cells. If ∆Ess < 0, mutated cell lines are sensitive to the inhibition of the drug target and wild-type cells are resistant; if ∆Ess > 0, wild-type cells are sensitive and mutated cells resistant. The rest of the columns are (% Mut Patient) percentage levels of each mutation in patients; (*p*-value) statistical significance before adjusting; and (lfdr) local false discovery rate adjusted *p*-value. Best predictive biomarker for each drug target is selected in this table (see complete data in Appendix A). Percentages of mutation in patients were downloaded from [35,36]. NA (Not Available) value indicates that there was no mutation data for that pair in [35,36]. * Targets with an existing approved or experimental inhibitor molecule.

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
