# Peer review of "DrugSniper, a Tool to Exploit Loss-Of-Function Screens, Identifies CREBBP as a Predictive Biomarker of VOLASERTIB in Small Cell Lung Carcinoma (SCLC)"

_cancers, 2020, doi:10.3390/cancers12071824_

Round 1
Reviewer 1 Report
The manuscript introduces readers to the DrugSniper tool, which can be used to link results of loss-of-function genetic screens with those from drug screening. The algorithm also interpolates information about oncogenic mutations in the underlying cell lines, such that it can make predictions about selective sensitivity to drug and gene perturbations. In principle, this is a novel, powerful tool to generate readily testable hypotheses related to synthetic lethality in cancer.
BUT, the major weakness is that while the authors present multiple examples of DrugSniper identifying connections between genes, drugs and biomarkers, this reviewer was only able to recapitulate results for PLK1 - CREBBP and PLK1-targeted agents. Every other example did not produce the reported results via DrugSniper. It's possible that the examples were simply not clearly explained, and thus it wasn't clear how to cue up analysis that would reproduce them. If that is the case, the authors should describe much more clearly the inputs relating to each example, such that the reader can produce the same outputs. If these examples are incorrect for any reason, they need to be removed from the manuscript.
If the only working example is the PLK1-CREBBP story, then the generalizability of the tool / workflow becomes questionable and the impact of the story greatly decreases. In that case, i would recommend against publication.
Reviewer 2 Report
Review Comments
The manuscript entitled “DrugSniper, a tool to exploit loss-of-function screens, identifies CREBBP as a predictive biomarker of Volasertib in small cell lung carcinoma (SCLC)” by Carazo et al describes genome-wide loss-of-function screens to model the sensitivity of inhibitors and predict biomarkers of sensitivity in tumor types. In this manuscript, authors have presented DrugSniper tool to identify response biomarkers for targeted therapies in 30 tumor types. However, the statements are exaggerated and needs further work to be used by researchers in the field. I do think the manuscript needs major revision before publication in Cancers.
To further improve the tool, I have few recommendations.
Comments:
- Authors presented tool DrugSniper by integrating loss-of-function screens data from Project Achilles, DRIVE and Score, and Cancer Cell Line Encyclopedia omics data. There are wide papers using this resource for personalized approach however it’s difficult to understand how authors attributed cell line data to patient data for patient centric approach (page 2 “stratification of patients based on the presence of biomarkers that predict the response to current therapies”). Currently authors have used cell line mutations with cell line sensitivity association approach to suggest personalized approach in patients which is exaggerated without applicable results.
- It is difficult to understand how the tool is unique and help researchers in the field to narrow down biomarkers in patients. Currently the tool identifies many biomarkers. The question remains how authors suggest to use this biomarker data. For example, in SCLC CREBBP mutations have 3 candidate biomarkers apart from PLK1.
- For generating database, authors have included both approved and \ investigational drugs targeted to protein inhibition. It is helpful to add FDA approved drug annotations which is useful for researchers. Some of investigational/experimental drug like “Aconitate Ion” does not fit into drug-target inhibition. It is helpful to remove such categories to focus on approved target specific drugs for clinical use by users.
- Identification of biomarkers is need to explained. it is important to note that the score obtained from RNAi/CAS9 suggest the gene essentiality to cell line and interpreted carefully like resistant or sensitive. Authors used mutations in cell line and essentiality score to come up with differential result using package limma which overall represents mutant vs non-mutant. But it does not explain same gene RNAi/cas9 inhibition effect and mutant (say gain-of-function) effect in cell line leading to possible biomarkers.
- Volasertib and BI2356 experimental results looks promising, however SCLC have about 30% of CREBBP mutations. Thus, highly mutated gene biomarkers would have been interesting. Do authors see similar results with PLK1 altered RNA expression in CREBBP mutation in SCLC and other cancers? Also table have 4 CREBBP mutation biomarker specific targets and other non- CREBBP mutation biomarker which is confusing.
- The statement “We have developed DrugSniper, a pioneer computational approach that facilitates the design of personalized therapies based on the status of gene mutations” seems more exaggerated. Design of personalized therapies using result is not explained/described in the manuscript.
- In manuscript, authors used tumor subtype term. Tumor subtype is specific term used delineate the heterogeneity in same cancer type. May be authors trying to suggest histological types.
- Difficult to understand “However, some drug activity processes, e.g., kinetics or cell 326 transportation, cannot be modeled by this pipeline. The output of DrugSniper should be therefore 327 considered as a prior hypothesis that helps a posterior drug development project, which should be 328 validated and tested.”
- Figure-1B, drug panel shows NA. When tried the page is blank.
- For tumor histological types, sample size 1,2 does not provide useful information thus can be merged.
- Table 1, how authors obtained % Mutated Patient.
- Selecting all cell lines and All Protein Targets (343) & Drugs & Mutations shows blank page. In another case, EGFR mutation selected in breast cancer cell lines which does not show EGFR as mutational biomarker and specific target in list.
Reviewer 3 Report
The manuscript entitled “DrugSniper, a tool to exploit loss-of-function screens, 2 identifies CREBBP as a predictive biomarker of 3 Volasertib in small cell lung carcinoma (SCLC)” presented by Caranzo et al., describes an interesting bioinformatics tool (DrugSniper) aimed at predicting response biomarkers for targeted therapies in multiple tumor types.
In general, this work starts as a bioinformatics investigation but it also includes good quality validations both in silico and in vitro. Differently from similar paper in the field of bioinformatics, I really appreciate the authors efforts in measuring the reliability of their software by performing robust validation of their predictions by mean of cell biology experiments. This is not typical among bioinformatics works, where validation of predictions is generally made by using literature data, and denote a certain grade of commitment of these authors in the cancer field.
The software is organized as both a web-server application which is freely accessible by researchers and a standalone version that can be downloaded from GitLab (a state of the art repository for software distribution) and run locally following the included instructions. Both the software versions are well conceived and work properly. The software use data retrieved by primary sources such as CCLE and Projects Achilles that should ensure a good coverage of the different cancer types.
Rather than simply present this new pipeline, the authors also used their software to investigate and identify novel protein-targeted for the treatment of small cell lung carcinoma (SCLC). Their analysis identified a remarkable vulnerability to polo-like kinase 1 (PLK1) inhibition in CREBBP-mutant SCLC cells. Validation were performed using cell viability assays, drug sensitivity assay and colony formation assay.
To the humble opinion of this reviewer, this work is very well conceived, developed and presented. This work is of a great interest as it represents a really smart utilization of data derived by large scale analysis and mining of available databases to propose new promising therapeutic approaches for contrasting cancer by repurposing existing drugs.
This tool will be very helpful to many researchers working on cancer.
Minor comments
1) Using the web application sometimes the cell lines are incorrectly reported. For instance, in tab “Select Cell Lines” marking Kidney in the left column and deselecting “Renal cell carcinoma” and “Renal cell carcinoma – Clear cells” the remaining option “Renal cell carcinoma – Papillary” is marked as containing data but them are not shown in the central box where in fact is reported “Slected Cell Lines (n=0). A similar situation with data apparently present but not shown in the central box appears to also involve the selection “Prostate”. I suggest the authors to verify if this behavior is expected or due to errors in programming the web application.
2) In vitro validation was conducted using four CREBBP mutated cell lines, however few details about the type of CREBBP mutations are given. How you selected these specific mutant cell lines?
3) I suggest the authors to include a link to the homepage directly in the main interface. Indeed, on the upper left corner the name of the application is available, however there is no way to go back to the main page after started the analysis (actually there is a link but it requires to click on “Bio Tecnun Apps” then select DrugSniper which is almost annoying). I suggest to modify “DrugSniper” tab to include the home page link.
4) Do you have an estimation of the software reliability when it is used as “no prior hypotheses”? (forth utilization case).
Reviewer 4 Report
The authors present a bioinformatics software tool called DrugSniper to predict the drug target/biomarker relations using loss-of-function screens on cell lines coupled with their molecular profiles. This tool also utilizes the drug databases such as DrugBank to map the identified targets with approved or investigational drugs. They present some results replicating the current knowledge of drug-target-biomarker groups as a proof of concept. They applied this tool to SCLC cell lines blindly and identified target-biomarker couples, such as PLK1-CREBBP, which is a known relation in various cancer types including NSCLC.
The statistical approach presented in this tool to map mutation profiles with targets is simply a t-test with multiple correction. Moreover, the tool is limited with the data available to in CCLE with 30 cell lines and their mutation profiles. Having said these, this tools still might be valuable as an over-the-shelf/easy-to-use tool for some researchers.
Below are my detailed comments:
Major Comments:
- Page 6, lines 179-183: More details about the findings for BRAF, MEK and ATM should be listed. For instance, BRAF which use case scenario was used, i.e. was the compounds input or was there a blinded search on the disease cohort? Also as in the case of BRAF, not all BRAF mutations are activating. BRAF V600E mutation causes the sensitivity to BRAF inhibitors. How does the software handle, if it does, such distinctions in identifying the biomarkers?
- Page 6, line 186: Similar to point made above, not all activating EGFR mutations are the same. For instance, kinase domain mutations vs extra-cellular domain mutations have very different drug response Was this distinction considered?
- Page 6, line 190: The finding of TP53 mutation as a resistance introducing factor for EGFR inhibitors is an important aspect of the presented tool, e. also resolving mutational co-occurrence patterns to address resistance. Authors should expand the method section on this case. If EGFR mutation-EGFR-Erlotinib relation was identified, how can this tool, with the form presented resolve the resistance introduced with co-mutation? Or was this an extrapolation of the results?
- Line 201: The parameter choice for percent of cell lines with essentiality score seem to be an important decision. Does this mean that you are looking for a genomic subtype with prevalence > 20%? More discussion/clarification is warranted on this choice.
- Line 206: As has been pointed above, the details of the mutational profiles, how the data is filtered and interpreted, which assumptions were made should be detailed as it is the sole of biomarker identification. Only filtering conditions were listed in the corresponding Figure.
- Any potential for using other data resources for either of the drug targets, target identification databases with mutation profiles should be mentioned. Is the tool scalable and flexible?
Minor Comments:
- Lines 351-353: The sentence has repetition
Round 2
Reviewer 1 Report
This reviewer thanks the authors for their responsive comments in the "response to reviewers" material. The information shared there greatly clarified why I was unable to produce the results described in Section 2.2 of the manuscript. I am now able to recapitulate some, but still not all of the examples. I believe that this is lack of familiarity with the DrugSniper tool as well as some instances of not enough detail being included in the Response.
I would like to see the authors include in the manuscript the section from page 2 of their Response that describes the examples as being "Visualize[d] case-by-case". In particular, text should be added to Section 2.2 from the Response that reads, "[For] pairs mentioned in the manuscript, ...we were only focusing on the statistical significance between the pairs. [These] were validated in the tab “Visualize case-by-case”. Some of them did not appear in the current pipeline due to the stringent filters e.g. we set a DEMETER Essentiality cutoff equal to -2 and a Delta Essentiality bigger than -2. We set these filters to minimize the false positives [when screening], which is crucial for wet lab hypothesis validation. These filters eliminated some of the well-known and clinically used pairs mentioned here, [although these] are well-known positives and [were statistically significant when assessed as pairs (case-by-case)]."
Much of the additional text with details on the precise analyses run and the box plots that visualize those results should be included in the Supplementary Materials.
Reviewer 2 Report
Please accept.
Author Response
Thank you for your review and useful comments
Reviewer 4 Report
The authors addressed all the questions and concerns adequately.
Author Response

(The authors gave the same response as above.)
